# Parathyroid Hormone (PTH)-Related Peptides Family: An Intriguing Role in the Central Nervous System

**DOI:** 10.3390/jpm13050714

**Published:** 2023-04-24

**Authors:** Cristina Dettori, Francesca Ronca, Marco Scalese, Federica Saponaro

**Affiliations:** 1Biochemistry Laboratory, Department of Pathology, University of Pisa, 56126 Pisa, Italy; 2Institute of Clinical Physiology, National Council of Research, 56126 Pisa, Italy

**Keywords:** PTH, PTHrP, PTH receptors, TIP39, PTH1R, PTH2R

## Abstract

Parathyroid Hormone (PTH) plays a crucial role in the maintenance of calcium homeostasis directly acting on bone and kidneys and indirectly on the intestine. However, a large family of PTH-related peptides exists that exerts other physiological effects on different tissues and organs, such as the Central Nervous System (CNS). In humans, PTH-related peptides are Parathyroid Hormone (PTH), PTH-like hormones (PTHrP and PTHLH), and tuberoinfundibular peptide of 39 (TIP39 or PTH2). With different affinities, these ligands can bind parathyroid receptor type 1 (PTH1R) and type 2 (PTH2R), which are part of the type II G-protein-coupled-receptors (GPCRs) family. The PTH/PTHrP/PTH1R system has been found to be expressed in many areas of the brain (hippocampus, amygdala, hypothalamus, caudate nucleus, corpus callosum, subthalamic nucleus, thalamus, substantia nigra, cerebellum), and literature data suggest the system exercises a protective action against neuroinflammation and neurodegeneration, with positive effects on memory and hyperalgesia. TIP39 is a small peptide belonging to the PTH-related family with a high affinity for PTH2R in the CNS. The TIP39/PTH2R system has been proposed to mediate many regulatory and functional roles in the brain and to modulate auditory, nociceptive, and sexual maturation functions. This review aims to summarize the knowledge of PTH-related peptides distribution and functions in the CNS and to highlight the gaps that still need to be filled.

## 1. Introduction

The appearance of bone tissue in vertebrates during evolution has been considered a crucial milestone, which is characterized by the development of a complex hormonal system in charge of mineral control, with a pivotal role played by the Parathyroid Hormone (PTH) [1,2].

Although PTH was classically understood as a principal modulator of calcium and bone metabolism, this view turned out to be simplistic, since an entire family of PTH-related hormones exists, with possible actions on other tissues and targets. In mammals, the known PTH-related peptides are Parathyroid Hormone (PTH), PTH-like hormones (namely, PTHrP and PTHLH), and tuberoinfundibular peptide of 39, namely, TIP39 or PTH2 [3]. Three genes are currently known to encode for the PTH family of hormones, and two genes code for two different receptors: parathyroid receptor type 1 (PTH1R) and type 2 (PTH2R) [4]. A third PTH receptor has been discovered in zebrafish and named PTH3R [5], but no human analog has yet been described. PTH-related ligands share seven to nine identical amino acid sequences, but this homology rises to 50% if we consider the number of amino acids and the size and charge of the whole molecule. Moreover, PTH1R and PTH2R show 51% residue identity, and can bind all the ligands, even if with different levels of affinity [6]. Both receptors are part of the type II G-protein-coupled-receptors (GPCR’s) family and are distributed in many organs and tissues other than bone, presumably mediating different physiological functions [7]. Early in the 1990s, some studies suggested a very intriguing role for PTH and PTH-related peptides in the Central Nervous System (CNS), since these ligands and their receptors are largely expressed in many areas of the brain [8]. However, for many years, few studies have explored the mechanism of PTH’s effects on the brain.

More recently, the clinical evidence of neuropsychological symptoms in patients affected by idiopathic and post-surgical hypoparathyroidism [9,10], lacking systemic PTH effects and the recently available human-recombinant PTH-substitutive therapy, has rekindled the interest in the possible effects of PTH and/or its related peptides on the CNS.

The aim of this review is to recapitulate the knowledge and the gaps in the literature regarding PTH-related hormones and their actions on the CNS.

## 2. PTH and PTH Type 1 Receptor: Role and Signaling Pathways

PTH is secreted by the chief cells of the parathyroid glands as a pre-prohormone, and subsequently cleaved in two steps within the cells to the active peptide of 84 amino acids. This is rapidly released by exocytosis in response to hypocalcaemic stimulus, and it is the major circulating and active form of PTH, even if other mainly inactive fragments are present in blood and cleared by the kidneys [11,12]. The active PTH 1-84 exerts a central role in the calcium/phosphorus homeostasis, directly acting on bone and the kidneys and indirectly on the intestine. In bone, PTH stimulates the immediate release of calcium from the skeletal store and stimulates bone resorption under a tonic secretion, whereas an intermittent stimulation leads to bone formation. In the kidney, the action of PTH is exerted on the distal tubule, where PTH stimulates calcium reabsorption, reducing urinary calcium excretion. In the intestine, PTH acts indirectly, promoting the final renal hydroxylation of 1,25(OH)_2_Vitamin D, which in turn increases the intestinal calcium absorption.

The regulation of bone metabolism by PTH is mediated by its binding to PTH1R, which is part of the GPCRs family and contains a long N-terminal extracellular domain, a seven-transmembrane domain, and a C-terminal tail extending in the cell. The 1–34 portion of PTH is able to bind the receptor in a “two-domain” model, which leads to G protein activation [11,13]. Different biochemical pathways are triggered by PTH–PTH1R binding: (1) when Gαs is activated, the signaling path of adenylate cyclase (AC) is switched on, followed by the downstream cyclic adenosine monophosphate (cAMP) and protein kinase A (PKA) cascade [14]; (2) another pathway can be alternatively activated when Gαq is recruited and the phospholipase C (PLC), protein kinase C (PKC) and inositol 1,4,5 triphosphate (IP3) path are turned on, which converges to mobilize calcium; (3) the PTH–PTH1R system can also recruit the pathway of β-arrestin, which ultimately activates the mitogen-activated protein kinase (MAPK) and other crucial cellular signaling molecules.

In bone, PTH–PTH1R binding mainly stimulates the PKA pathway to mediate both the anabolic and catabolic effects of the hormone, even if the PKC cascade also has a role. PTH1R is present in cells of the whole osteoblastic lineage, and PTH stimulates the differentiation of progenitors and the proliferation and activation of osteoblasts, ultimately inducing bone mineralization. On the other hand, PTH–PTH1R binding is also responsible for a “dual” effect on bone, namely, the coupling of bone formation with bone resorption via the RANK-RANK Ligand system. The tumor necrosis factor-related cytokine receptor- activator of the nuclear factor K-B (RANK) ligand (RANKL) is produced by osteoblasts under the effect of PTH, and it is a crucial enhancer of the recruitment of osteoclasts and the increase in bone resorption. 

The classic effects of PTH–PTH1R on bone metabolism are well known. However, PTH1R’s wide distribution in organs other than bone and kidneys suggests a more complex and still poorly investigated systemic effect of the PTH system, which reaches the CNS via direct and indirect actions.

## 3. What Do We Know about PTH–PTH1R in the Brain?

The intuition of one effect of PTH on the CNS is derived from clinical evidence. Systemic diseases involving increased PTH secretion, such as Hyperparathyroidism, as well as PTH deficiencies such as Hypoparathyroidism, are accompanied by a variety of neuropsychological symptoms and cognitive and affective impairments, suggesting that PTH could also exercize effects on the brain. Nevertheless, after initial enthusiasm about the search for PTH and PTH1R distribution in the brain, the literature in this field has slowed down, and the molecular mechanisms involved are still unclear and partially unexplored. We here present evidence for the existence of PTH–PTH1R in the brain and the possible associated effects.

### 3.1. Distribution of PTH/PTHrP-PTH1R in CNS

PTH mRNA in the brain was discovered in 1985, when Balabanova et al. first studied the presence of immunoreactive PTH in the brain of sheep. They demonstrated PTH and enzymes degrading PTH in many areas of the brain and in the pituitary tissue of 21 animals [15]. Radiolabeled PTH has also been found to bind to the membranes in the hypothalamus, and with a lower affinity in the cortex and cerebellum of rats’ and rabbits’ brain [16]. Controversial and old data are present in the literature regarding PTH crossing the blood–brain-barrier (BBB) [16]. PTH has been found in the cerebrospinal fluid (CSF) [17], and a recent review summarized data about the association between changes in the levels of PTH in the CSF and the incidence of neuropsychiatric disorders in patients [18]. An intriguing but so far undemonstrated hypothesis postulates that a small intracerebral secretion of PTH could occur due to the similarities between parathyroid cells and neuroectodermally-derived APUD cells in the brain, and the sequence homology between PTH and the neuropeptide proopiomelanocortin (POMC) [19].

More evidence is present in the literature regarding the distribution and expression of PTH receptors in several areas of the CNS. In humans, PTH1R is known to be expressed in the hippocampus, amygdala, hypothalamus, caudate nucleus, corpus callosum, subthalamic nucleus, thalamus, substantia nigra, and cerebellar astrocytes. Eggenberger et al. cloned and functionally characterized the PTH1R from the human cerebellum, which was identical to the PTH1R expressed in the kidney and in bone cells. The distribution of the mRNA of the receptor in the brain was evaluated by Northern Blot analysis and In Situ Hybridization Histochemistry performed on brain tissue sections of rats and humans. Moreover, functional analysis after the expression of the receptor in the neuroblastoma SK-N-MC cell line confirmed the same functional properties of the kidney receptor [20]. Weaver et al. confirmed a widespread but anatomically related distribution of PTH1R mRNA in the brains of adult rats via In Situ Hybridization. The trigeminal nucleus and the trigeminal ganglion, the lateral reticular, the pontine and the reticulotegmental nuclei, the hypoglossal nucleus and the area postrema were the areas with the highest expressions of PTH1R, but it was found also in the amygdala, cerebellum and cortex [21]. More recently, Evliyaoglu et al. studied the PTH1R receptor via Western Blot analysis and Immunohistochemistry in samples from normal human brains and 73 glial tumors with different degrees of aggressiveness; the receptor was present both in non-neoplastic tissue (neurons, astrocytes, and endothelial cells) and in neoplastic tissue, but was more highly expressed in pure human astrocytic tumors [22].

Similarly to PTH, PTHrP has also been found in many regions of the CNS, such as the hippocampus, hypothalamus, pituitary and cortex, and has been measured in the CSF [23]. In vitro studies suggest that PTHrP could play a role in protecting neurons from prolonged depolarization toxicity, the so called “excitotoxicity” issue [24]. This hypothesis seems to have been confirmed in in vivo studies in which PTHrP-/- knockout mice were shown to be very sensitive to kainate-induced seizure [24]. Moreover, PTHrP is also expressed in glia and astrocytes, where it seems to be involved in inflammatory responses to brain injury in animal models [25]. Other data suggest its possible role in the modulation of nerve regeneration [26].

### 3.2. Effects of the PTH/PTHrP-PTH1R System on the CNS

Despite the evidence of PTH in the brain, the local effects of the hormone have been poorly elucidated. In the 1980s, direct actions of PTH in the brain were demonstrated on memory and learning processes: the administration of intact PTH and fragmented PTH_44-68_ in the lateral ventricle of rats improved the capacity of shuttle-box active avoidance, ameliorated memory [27], and reversed the catalepsy pharmacologically induced by haloperidol [16]. A recent study by Chen et al. tried to elucidate the mechanisms underlying the protective effect of PTH on memory using 5XFAD mice, a well-characterized model of Alzheimer’s Disease (AD). They treated 5XFAD mice and controls with PTH analogue PTH_1-34_ daily for 3 months and assessed brain pathology and behavior. They observed an improvement in memory and learning via behavioral testing (Novel Object Recognition (NOR) and Morris Water Maze (MW)) in those treated with PTH_1-34_ compared to controls, with a major effect noted in females. They demonstrated a reduction in plague-associated dystrophic neurites in the treated 5XFAD, which is a marker of neurodegeneration, and a significant reduction in neuroinflammation markers and astrocyte apoptosis. These data suggest that PTH action in the brain could be directed towards brain astrocytes, suppressing cellular death and neuroinflammation [28].

Some studies have provided evidence of the neuroprotective role of PTH and PTHrP through a modulation of the cerebrovascular system. PTH can enhance neuroangiogenesis and neuroblast migration in ischemic cortical tissue, but it is unclear if this is a direct local action or an indirect systemic effect, and which is the mediator [29]. More solid data have been published regarding the role of PTHrP as a modulator of cerebral vasculature, which can be summarized as follows: PTHrP is secreted in the endothelium of the cerebral microvasculature after brain ischemia; the administration of exogenous PTHrP enhances vasodilation in an in vitro model of cerebral vasculature; and the administration of PTHrP reduced the size of a brain infraction in an in vivo model of rats [30,31]. Taken together, these data prompt further studies that will shed light on the intriguing action of PTH/PTHrP as potentially neuroprotective molecules.

Another mechanism proposed for the local action of PTHs in the brain is the paracrine modulation of the catecholamine metabolism. Indeed, the intracerebral administration of PTH increased glutamic acid decarboxylase activity action in the substantia nigra [32], and Harvey et al. demonstrated that the intracerebroventricular injection of PTH, either human or rat, was able to specifically increase the DOPAC (dihydroxyacetic acid) and DOPAC/dopamine ratios in the hypothalamus [33].

Hyperalgesia modulation has been proposed as another effect of PTH in the brain, as advocated in the 1980s by Gennari et al. [34]. This hypothesis was confirmed by a recent study in which the effect of the PTH analogue teriparatide (PTH_1-34_) on sensory neurons in ovariectomized (OVX) rats was evaluated. PTH_1-34_ was able significantly and rapidly to reduce thermal hyperalgesia in OVX rats after administration [35]. These data suggest that PTH_1-34_ could pharmacologically act on central neurons and could explain the clinical evidence that patients with osteoporosis treated with PTH_1-34_ experience fast and sensitive relief of back pain [36,37].

## 4. PTH2R and Its Ligands: Expression and Role in the Brain

### 4.1. PTH2R Structure and Functions

During investigations for novel type II GPCR’s, in 1995, Usdin et al. identified a second PTH receptor, which was named PTH2R. The major unexpected result from their initial study was that PTH2R was able to discriminate between PTH and PTHrP: indeed, in the COS-7 cell line, PTH caused a 5–25-fold increase in cAMP over the basal level, while no effect was obtained after PTHrP administration [4]. The low expression of mRNA associated with PTH2R was demonstrated in human bone cells and kidney cells, but it was highly expressed in the pancreas and brain, suggesting a role in metabolism and in the CNS [4,38]. Subsequently, it was demonstrated that the PTH2R in rats is activated by PTH-related peptides (PTH_1-84_, PTH_1-34_) with much lower potency than the human receptor; this observation led to the search for different ligands for PTH2R, and TIP39 was discovered, as discussed in detail in the next paragraph [39]. Therefore, in humans, PTH2R is likely to mediate the central effects of PTH-related peptides, be they either PTH or TIP39, with modality and effects remaining unclear.

PTH2R’s signaling pathways have been studied and partially elucidated: the activation of PTH2R leads to cAMP production, similarly to PTH1R, and also induces an increase in [Ca2+]i, which rapidly declines after 80 s. The cAMP pathway seems to be preferred, and the receptor’s modality of activation depends on the ligand. The response to PTH is very quick; in contrast, the response to TIP39 is prolonged and sustained [40,41]. These data indicate different roles for PTH and TIP39 in modulating the PTH2R response in humans.

The presence of PTH2R in humans was clarified by Bagò et al. in 2008; RNA from post-mortem human cortical and brainstem tissue was isolated and reverse-transcribed for RT-PCR, demonstrating PTH2R mRNA in the brain. Moreover, fluorescent Immunochemistry [42] found that PTH2R immunoreactive fibers were densely present in a human medulla oblonga. This localization had already been demonstrated in rodent tissues, where it was correlated with the pronociceptive function of the receptor, suggesting a similar distribution of PTH2R in humans and mice/rats [43]. Co-localization studies and anatomical data from the same group have suggested a possible involvement of the PTH2R system in the regulation of stress response, fear, reproductive behavior, and nociception modulation in primates and rodents [44]. More recently, Gellen et al. evaluated the involvement of PTH2R in maternal mice behavior. They evaluated body temperature (Tc) changes, depression-like behavior, and locomotor activity in PTH2R knockout (PTH2R-KO) mice compared to the wild type; PTH2R-KO female mother mice showed a significantly lower Tc, a higher locomotor activity, and higher depressive moods under different settings of behavioral testing, suggesting a possible role of the PTH2R system in maternal behaviors [45].

### 4.2. TIP39/PTH2R System in Brain

The tuberoinfundibular peptide of 39 residues TIP39 is a small peptide belonging to the PTH-related peptides family, and it is able to specifically bind PTH2R [46].

TIP39 was first extracted from bovine hypothalamus samples in 1997, and its chemical structure was subsequently clarified by NMR spectroscopy [47].

The primary sequence of the peptide is not very similar to that of PTH, but its 3D backbone is surprisingly similar. Indeed, TIP39 presents an N-terminal and a C-terminal alpha helix separated by a flexible central region, and the two peptides show high sequence homology in the alpha helix of the N-terminal region, thus explaining the similarity in the signaling pathway. Nevertheless, there are great differences in the flexible central region and in the alpha helix of the C-terminal region, where the presence of some charged amino acids in the hydrophobic portion of the helix changes the level of amphipathicity, which is lower in TIP39 than in PTH. In addition, further amino acid differences in certain key regions lead to the higher affinity of TIP39 for PTH2R than with PTH [48,49].

TIP39 was initially purified and studied in HEK293 cells after transfection with PTH2R based on cAMP production. These studies showed that TIP39 is a natural ligand of PTH2R, since TIP39 promotes a strong response, and increases both cAMP and Ca^2+^ levels. The specificity and selectivity of PTH2R binding by TIP39 is due to the N-terminal part of the peptide, since the removal of certain residues in this region leads to the loss of PTH2R activation [13,50,51]. Further studies with HEK293 and COS-7 cell lines confirmed that TIP39 can bind PTH1R with moderate affinity (~60 nM), but it seems not to activate the receptor [52]. Interestingly, when some residues from the N-terminal are removed, the affinity for PTH1R dramatically increases: TIP 7-39 binds PTH1R with an affinity of 17 nM, and it is the most powerful known PTH1R antagonist [53].

In humans, TIP39 is codified by a gene located on chromosome 19, long arm, 19q13.3, and contains two exons and one intron, similarly to the mouse gene. In Situ Hybridization and Immunohistochemical studies have shown that TIP39 is widely expressed within the body in regions such as the brain, liver, kidneys, and heart, in both fetal and adult life, with a specifical anatomical and temporary distribution [52,54].

Focusing on the brain, TIP39 is highly expressed in three regions of the subparafascicular area in the caudal thalamus, defined as the Periventricular Gray of the Thalamus (PVG), the Posterior Intralaminar Complex of the Thalamus (PIL) and the Medial Paralemniscal Nucleus (MPL) [55,56]

The PVG region highly expresses TIP39 in the adult rat brain [57,58]. The PIL region presents a high number of TIP39-expressing neurons during the embryonic and postnatal stages, which decrease during growth, compared to the PVG. Finally, there are many TIP39-expressing neurons in the MPL region, where they account for 75% of the cell population [59].

All these CNS regions are highly connected to the forebrain, diencephalon (such as hypothalamus and geniculate body), and the superior and inferior colliculus, cuneiform nucleus, medial paralemniscal nucleus, and parabrachial nuclei [60]. The PIL area receives projections from the medial prefrontal, insular and somatosensory cortices, the preoptic area, the hypothalamus, the amygdala, the tegmental area, the parabrachial nuclei, the solitary tract, and the trigeminal nucleus. The MPL has many neuronal connections with the auditory cortex, geniculate body, colliculus, external and dorsal cortices, hypothalamus, thalamic nuclei, preoptic area, and others [61].

The anatomical organization of TIP39 neurons is consistent with the distribution of PTH2R and with the proposed functional roles of the TIP39–PTH2R system in the regulation of limbic and endocrine functions, auditory and nociceptive processes, and sexual maturation [55,60,61]. Encouraged by localization and functional in vitro and in vivo studies, further animal studies have been carried out to evaluate the direct and indirect effects on behavioral actions. TIP39 itself was intraventricularly administered in rodent models; specific and non-specific TIP39 antagonists were developed, and transgenic knockout mice for TIP39 and/or PTH2R were studied [38,62,63].

The involvement of TIP39 in nociceptive functional control has been evaluated by comparing wild type mice and mice administered with a PTH2R antagonist, such as HYWH-TIP39 or TIP39 itself. TIP39 was able to decrease the latency in acute nociceptive sensitivity tests, while the antagonist increased the latency. In mice with nerve injury, PTH2R and/or TIP39 suppression led to reduced tactile and thermal hypersensitivity, compared to controls. All these data confirm a role for the PTH2R–TIP39 system in the modulation of nociception [64,65].

The PTH2R–TIP39 system also plays a role in the fear response, since the deletion of the system in knockout mice leads to a fear incubation effect, namely, a time-dependent increase in fear response [66,67]. The hypothesis is a both direct and indirect effect, mediated by the modulation of the neuroendocrine system and the regulation of stress hormone release [68].

TIP39 has displayed an anxiolytic and anti-depressive effect in mice injected with TIP39 into the lateral cerebral ventricle, using the elevated plus maze test and forced swim test [62]. Moreover, TIP39 and PTH2R knockout mice showed an increased anxiety-like behavior and increased immobility and freezing times [69].

Due to the abundance of TIP39 fibers and PTH2R expression in the preoptic area of the hypothalamus, which is crucial to maternal behavior in rodents, the effects of TIP39 on maternal attachment and motivation have been tested. The specific local administration of HYWY-TIP39, a potent antagonist of PTH2R, dramatically reduced pup-induced place-preference [70]. The hypothesis is that TIP39 modulates maternal behavior by acting on galanin preoptic neurons, which are abundantly innervated by TIP39-containing fibers [71].

These all data confirm that TIP39 plays a role in the CNS via a possible autocrine and/or paracrine mechanism, even if an endocrine function is suspected but has not yet been demonstrated.

## 5. PTH and Brain: Evidence from Clinical Settings

The possible role of PTH and PTH-related peptides in CNS is also suggested by data in clinical settings. Patients with both Primary Hyperparathyroidism and Hypoparathyroidism complain of neuropsychological and cognitive symptoms and show a low quality of life (QoL). These observations suggest a role for PTH, either when it is absent/insufficient, or when it is inappropriately elevated. Whether the effects on the CNS are due to a direct action of PTH on the brain, or are secondary to hypo- and hypercalcemia, has not yet been completely elucidated; however, it is reasonable to argue that at least some of these effects are mediated by PTH and PTH-related peptides, due to the amount of evidence previously summarized.

In patients with Hypoparathyroidism, the lack of PTH has a profound effect on calcium homeostasis, which is only partially restored by the classical treatment with calcium and vitamin D active analogues [72,73]. New perspectives have recently opened with the introduction of a novel substitutive therapy with human recombinant PTH, which is still not available for all patients [74,75]. In addition to the kidney and bone manifestations of Hypoparathyroidism, patients experience a real neuropsychological burden with neuropsychological and cognitive symptoms, including depression, fatigue, anxiety, and memory dysfunction [76]. Cognitive and affective symptoms have been evaluated as part of QoL, and patients with Hypoparathyroidism show an impairment in this, in both the mental and physical domains, compared to healthy controls and other patients. QoL has been evaluated by self-administered tests, such as the 36-Item Short Form Health Survey (SF-36) and the WHO5 Well-Being Index Survey (WHO-5); these tools are widely used and validated, but they have the disadvantage of not being specific to the disease [77,78,79]. More recently, Hypoparathyroid Patient Experience Scale–Symptom (HPES-Symptom), a more specific test, has been validated for Hypoparathyroidism, confirming the low QoL in patients with the disease [80]. Moreover, a novel neuropsychological approach, using expert neuropsychologist-devised tests, has allowed the analysis of cognitive, executive, and attentive functions in patients with Hypoparathyroidism. Patients with both idiopathic and post-surgical Hypoparathyroidism present neuropsychological impairment, with a worse performance in these tests for the former and for patients with longer follow-up and lower calcium levels, as evaluated in studies by our group and others [9,10,81].

On the other hand, the inappropriate increase in PTH in patients affected by Primary Hyperparathyroidism (PHPT) [82,83] is correlated with a variety of neuropsychological symptoms, ranging from depression, anxiety, and sleeping disorders to dementia, psychosis, and cognitive decline [84]. Most neuropsychological symptoms are described in symptomatic PHPT, which is generally related to severe hypercalcemia, but they are also present in asymptomatic PHPT, with a prevalence of up to 60% of patients in some settings [85,86]. Prospective case–control studies have shown cognitive impairment, reduction in verbal and non-verbal memory, and a deficit of attention and semantic fluency, as evaluated by specific neuropsychological tests [87,88]. A study in a large cohort of patients with both symptomatic and asymptomatic PHPT showed an impairment of the Mini-Mental State Examination test for 25% of patients, with a significant improvement after parathyroidectomy (PTx) [86]. The QoL in patients with PHPT is reduced, in both the mental and physical domains, and is significantly improved after PTx [89,90,91]. However, few and inconclusive data are currently available from randomized clinical trials, so the most recent Fifth International Workshop for PHPT did not include neuropsychological symptoms in the criteria for PTx [92].

In both Hypoparathyroidism and PHPT, PTH disturbances are correlated with neuropsychological symptoms, with a possible U-shaped curve. The possible pathophysiological mechanisms are related to the role of PTH in controlling calcium levels in the blood, but also to the direct effect of PTH in modulating calcium flux in the CNS, which is crucial for the adequate functioning of neurons [93]. Moreover, PTH has been described as modulating the cerebral capillary blood flow, in addition to its possible direct action on PTH1R and PTH2R in the brain.

## 6. Conclusions

In conclusion, evidence from the available literature confirms intriguing roles for PTH-related peptides in CNS, which are summarized in Table 1. However, the underlying mechanisms are still largely unknown, and the clinical implications encourage further studies in vitro, in vivo, and under clinical settings.

## Figures and Tables

**Table 1 jpm-13-00714-t001:** PTH-related peptides and their receptors in CNS.

PTH-Related Family Peptide	Receptor with Higher Affinity	Distribution in CNS	Possible Effects on CNS	Proposed Mechanisms
PTH	PTH1R	Hippocampus, amygdala, hypothalamus, caudate nucleus, corpus callosum, subthalamic nucleus, thalamus, substantia nigra, cerebellum	Improvement of memory and learning. Hyperalgesia modulation. Endocrine/paracrine function.	Protection of brain astrocytes by suppression of cell death and neuroinflammation. Modulation of catecholamine metabolism in brain. Modulation of cerebrovascular system, promoting neuroangiogenesis.
PTHrP	PTH1R	Cerebral cortex, hippocampus, cerebellum	Reduction of brain ischemia.	Action on endothelium of cerebral vasculature after brain ischemia, enhancement of vasodilation and neuroangiogenesis.
TIP39	PTH2R	Thalamus (PVG, PIL, MPL regions), amygdala, Locus coeruleus, Hypothalamus	Endocrine effects. Modulation of auditory response, nociception, fear response. Anti-depressive effects. Modulation of maternal behavior.	Regulation of neuroendocrine system in brain, modulation of cathecolamine, vasopressin, prolactin. Modulation of glutamate neurotransmitter.

PTH: Parathyroid Hormone; PTHrP: PTH related peptide; TIP39: tuberoinfundibular peptide of 39; PVG: Periventricular Grey of the Thalamus; PIL: Posterior Intralaminar Complex of the Thalamus; MPL: Medial Paralemniscal Nucleus.

## Data Availability

Not applicable.

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
