# Peer review of "Parathyroid Hormone (PTH)-Related Peptides Family: An Intriguing Role in the Central Nervous System"

_jpm, 2023, doi:10.3390/jpm13050714_

Round 1

Reviewer 1 Report

The manuscript describes the function of PTH, its related peptides and their receptors in the central nervous system. I consider that the manuscript summarizes important and interesting functions of this peptides family. As there are several paralogous peptides and receptors within this family, I think it is better to add a table of the peptides (PTH, PTHrP, TIP39) with their receptors (PTH1R, PTH2R) about the affinity, distribution, cell signaling, possible function to avoid confusion. Current Figure 1 is too simple and just listing different peptides, receptors, and functions. Readers cannot get an insight from this figure. It should be improved. Which peptide system is related to which function? Where do they exist in the brain? Its nothing to show a brain like cartoon in the middle.

Minor comment

line 21: characterized

line 64: exert

Author Response

We thank the reviewer for the suggestions: we deleted the figure which was not enough informative and we included in the revised version a Table, as suggested, which contains data regarding the peptides, their receptors, the distribution in CNS, effects and known/proposed mechanisms.

We tried to improve the manuscript and you will find the changes in red.

Reviewer 2 Report

The authors gave the overview about the role of parathyroid hormone (PHT), PTH related family peptides, along with appropriate receptors in central nervous system. The review is of sufficient significance and originality, however there are numerous issues that need to be addressed:

1                 In the title, the abbreviation PHT should be introduced/defined.

2             In the affiliation “1Biochemistry Laboratory, Department of Pathology, University of Pisa (Pisa, Italy), federica.saponaro@unipi.it” the addresses for corresponding author is listed.

3                 Section Abstract is too short, poorly fitted and confusing. For the example, what are the other physiological effects of PTH? What are the components of PHT-mediated pathways (ligands and receptors)? Which downstream molecules/signaling pathways of PHT receptors are affected? Or how the signal is further transmitted? Is oxidative stress involved? How are the components of PHT signaling pathways, including ligands and receptors associated with memory, hyperalgesia, nociception, fear response, anxiety and depression? Specify which specific PTH related peptides will be discussed in the review. It is necessary to introduce/defined the explanations for the used abbreviations along with appropriate/relevant information. These are just some examples of the shortcomings of this section since it should be much more fluent and informative.

4               All other sections are, also, very confusing, not thorough enough and the key points are not well discussed. The provided explanations/information of the listed studies are not thorough enough. The description of PHT-mediated signaling in more details is missing, which signaling pathways are involved and how, etc. Also, the information whether PHT and PHT-related peptides, or PHT-mediated signaling affect the cerebrovascular system is lacking. Could alterations in concentrations of PHT and PHT-related peptides or PHT-mediated signaling be associated with cerebrovascular/neurodegenerative diseases/conditions? These are just some of the examples of the shortcomings of these sections and the manuscript in general. Thus, the manuscript is not informative enough and fluent and it should be improved and written with more details from the literature that is relevant for the topic of the review.

5              The listed literature is outdated. Moreover, the list of references is too big for the amount of data incorporated into the manuscript.

6               The English is very difficult to understand and it is necessary to check the spelling and grammar throughout the manuscript since there are numerous errors.

Author Response

We thank the reviewer for the detailed revision, we changed our paper accordingly to the suggestions and we think that the effort produced an improved manuscript.

  1. The abbreviation has been defined
  2. The address has been removed
  3. The abstract has been completely changed accordingly to suggestion.
  4.  The whole manuscript has been revised to be clear and detailed and to provide explanation/information about the cited studies. We changed the paragraphs and we dedicated a section to PTH mediated signaling pathways in details as required (section 2). Moreover we dedicated a paragraph to the known effects of PTH/PTH1R in CNS, with more recent studies included in the revised version, even if we recognize that many signaling patways are still unknown (section 3.2). Moreover, we included the potential effects of PTH and PTHrP on cerebrovascular system as suggested and some associations with cerebrovascular conditions. All changes are highlighted in red.
  5. The bibliography has been updated and the amount of data incorporated has been improved.
  6. The manuscript underwent a certified English revision by MDPI English service and also to accurated plagiarism ceck.

Round 2

Reviewer 1 Report

I consider that the authors have adequately revised the manuscript.

Reviewer 2 Report

The authors have addressed the most of the concerns. However, I have still noticed some issues with the English language and style.